# Monitoring of mitochondrial oxygen tension in the operating theatre: An observational study with the novel COMET® monitor

**Floor A. Harms** [ORCID]\*[©], **Lucia W. J. M. Streng**[©], **Mark A. Wefers Bettink**, **Calvin J. de Wijs**, **Luuk H. Römers**, **Rineke Janse**, **Robert J. Stolker**, **Egbert G. Mik**

Department of Anesthesiology, Laboratory of Experimental Anesthesiology, Erasmus MC, University Medical Center Rotterdam, Rotterdam, The Netherlands

[©] These authors contributed equally to this work.

\* f.harms@erasmusmc.nl

**Data Availability Statement:** The data is made available on Dryad: https://doi.org/10.5061/dryad. vdncjsxzr.

## Abstract

### Introduction

The newly introduced Cellular Oxygen METabolism (COMET®) monitor enables the measurement of mitochondrial oxygen tension (mitoPO$_2$) using the protoporphyrin IX triplet state lifetime technique (PpIX-TSLT). This study aims to investigate the feasibility and applicability of the COMET® measurements in the operating theatre and study the behavior of the new parameter mitoPO2 during stable operating conditions.

### Methods

In this observational study mitochondrial oxygenation was measured in 20 patients during neurosurgical procedures using the COMET® device. Tissue oxygenation and local blood flow were measured by the Oxygen to See (O2C). Primary outcomes included mitoPO$_2$, skin temperature, mean arterial blood pressure, local blood flow and tissue oxygenation.

### Results

All patients remained hemodynamically stable during surgery. Mean baseline mitoPO$_2$ was 60 ± 19 mmHg (mean ± SD) and mean mitoPO$_2$ remained between 40–60 mmHg during surgery, but tended to decrease over time in line with increasing skin temperature.

### Conclusion

This study presents the feasibility of mitochondrial oxygenation measurements as measured by the COMET® monitor in the operating theatre and shows the parameter mitoPO$_2$ to behave in a stable and predictable way in the absence of notable hemodynamic alterations. The results provide a solid base for further research into the added value of mitochondrial oxygenation measurements in the perioperative trajectory.

**Funding:** The authors received no specific funding for this work.

**Competing interests:** Dr. Mik is listed as (co-) inventor on the following patents and patent applications, held by the Academic Medical Center in Amsterdam or the Erasmus Medical Center in Rotterdam, The Netherlands. European patent EP09788277.3 "Methods and devices for assessment of mitochondrial function". US Patent 13/059,225 "Methods and devices for assessment of mitochondrial function". European patent EP1904831 "Device and method for determining the oxygen concentration of a substance". US Patent 8,008,038 "Methods for determining oxygen concentration with protoporphyrin IX". International patent application PCT/EP2021/063642 "A method and device for optical quantification of oxygen partial pressure in biological tissue". The patents are licensed to Photonics Healthcare B.V., Utrecht, The Netherlands. Photonics Healthcare is the developer and manufacturer of the COMET device. Dr. Mik is founder and shareholder of Photonics Healthcare. This does not alter our adherence to PLOS ONE policies on sharing data and materials. The remaining authors declare no competing interests.

**Abbreviations:** ALA, 5-aminolevulinic acid; ADP, adenosine diphosphate; AMP, adenosine monophosphate; ATP, adenosine triphosphate; AU, arbitrary unit; COMET, Cellular Oxygen METabolism; CVP, central venous pressure; $FiO_2$, inspired oxygen fraction; IRB, Institutional Research Board; IV, intravenous; MAP, mean arterial pressure; $MitoPO_2$, Mitochondrial oxygen tension; O2C, Oxygen to See; Pi, inorganic phosphate group; PpIX-TSLT, protoporphyrin IX triple state lifetime technique; RDM, rectangular distribution method; SD, standard deviation; SNR, signal/noise ratio; SpHb, continuous total hemoglobin; $StO_2$, capillary venous saturation; TIA, transient ischemic attack; VU, velocity units.

## Introduction

Safeguarding adequate tissue oxygenation to ensure functional aerobic metabolism is an important task of the anesthesiologist during general anesthesia. The oxygen pathway from air to the mitochondria starts with the inhalation of air into the lungs. In the lungs it diffuses through the alveoli into the circulating blood where it binds to hemoglobin. Oxygen is then delivered to tissue cells via macro- and microcirculatory flow and finally by the diffusion of molecular oxygen. Eventually the oxygen molecule reaches its ultimate destination, the mitochondria. Here oxygen is used in the oxidative phosphorylation process to produce adenosine triphosphate (ATP) that acts as the main energy source for many cellular processes.

Current clinical management for maintaining adequate tissue oxygenation is mainly focused on the normalization of systemic hemodynamic parameters such as blood pressure (BP), heart rate (HR), hemoglobin levels, peripheral oxygen saturation, cardiac output and central venous saturation. However, a significant limitation of these monitoring techniques is that they are poor indicators of early tissue hypoxia [1, 2]. Moreover, the role of lactate as a measure of local oxygen delivery has been brought up for debate. As multiple pathways can cause an increased serum lactate besides just anaerobic metabolism induced by tissue hypoxia [1, 3, 4]. Therefore, new methods must be developed and analyzed to monitor tissue oxygenation.

Over the last couple of decades several novel measurement techniques have been developed to monitor tissue $PO_2$. Amongst others, these include; near-infra-red spectroscopy (NIRS) [5], side-stream dark field imaging of the microcirculation [6], and the monitoring of tissue oxygen tension of subcutaneous tissue [7] or conjunctiva [8]. None of these techniques have had widespread success or have been integrated in routine clinical practice, as a result of their complexity, and reliability- and technical issues [9]. With the introduction of the Cellular Oxygen METabolism (COMET®) device we developed a monitoring technique that aims to overcome the afore mentioned hurdles [10].

The COMET device is based on delayed fluorescence of protoporphyrin IX using the triplet state lifetime technique (PpIX-TSLT) [11–14]. PpIX-TSLT measurements are quantitative and have been calibrated in situ, using isolated cells and isolated organs [11, 15–17]. The calibration constants were found to be identical for several organs and have also been validated in the human epidermis [18–20]. After, the promising results of a feasibility study in healthy human volunteers the clinical measuring device was developed and named the COMET® [10, 14]. Since the introduction of COMET® it has been utilized in several clinical studies. To date, the COMET® has been used in studies performed with critically ill patients [21, 22], healthy volunteers [23] and surgical newborns [24]. Additionally, Ubbink [10] and Harms [25] et al have presented several case reports wherein mitochondrial oxygenation ($mitoPO_2$) measurements are described amid hemodynamically unstable surgical situations.

To further build on the pre-existing research, this observational pilot study focused on the feasibility and performance of the COMET® under real-life circumstances during surgical procedures. The primary aim was to study the intra-operative behavior of $mitoPO_2$ during stable hemodynamic circumstances. The hypothesis was that $mitoPO_2$ would remain stable during hemodynamically stable conditions during surgery. The secondary aim was to identify potential influences on the measurements which could have been overlooked using animal models or healthy volunteers. The study population comprised of a cohort of neurosurgery patients, since it is a relatively healthy patient population, with stable operation conditions and a relatively long surgery time.

## Materials and methods

### Trial design, setting, participants

The study was approved by the Institutional Review Board (IRB) at the Erasmus Medical Center CCMO-register 'Non-invasive monitoring of mitochondrial oxygen consumption and oxygenation (COMET): observational clinical study', (NL51937.078.15). All study procedures were performed in accordance with the relevant guidelines and regulations.

This single center observational study was performed at the Erasmus Medical Center, a tertiary care center in the Netherlands. Inclusions ran from April 2016 up until March 2020. Patients aged between 18–70 years old and scheduled for neurosurgery were considered eligible for participation. Exclusion criteria included surgeries without the need for invasive intra-arterial blood pressure monitoring, presence of mitochondrial diseases and pregnancy or lactation. The goal of this study was to measure the mitoPO$_2$ intraoperatively under stable hemodynamic conditions. Therefore, patients were excluded if there was hemodynamic instability during surgery namely, if blood transfusion was required, if the mean arterial pressure (MAP) decreased more than 25% from baseline or if high doses of vasopressors were needed (noradrenaline $\geq$ 0,10 mcg/kg/min or the equivalent dose of phenylephrine). According to standard protocol, hypertension was medically induced during the hemostasis phase. This raise in blood pressure was not included in the analysis of the results.

Patients were screened by the research team, based on data collected at the pre-operative consultation. Written informed consent was obtained from all patients prior to study participation.

### Principle of mitoPO$_2$ measurements

The background and principles of the PpIX-TSLT are described in detail elsewhere [10, 11, 17]. In short, PpIX is the final precursor of heme in the heme biosynthetic pathway. PpIX is synthesized in the mitochondria, and administration of 5-aminolevulinic acid (ALA) substantially enhances the PpIX concentration. PpIX possesses a triplet state that reacts strongly with oxygen, making its lifetime oxygen-dependent. Population of the first excited triplet state occurs upon photo-excitation with a pulse of light, and causes the emission of red delayed fluorescence. The delayed fluorescence lifetime is related to mitoPO$_2$ according to the Stern-Volmer equation:

$$mitoPO_2 = \frac{\frac{1}{\tau} - \frac{1}{\tau_0}}{k_q}$$

in which $\tau$ is the measured delayed fluorescence lifetime, $k_q$ is the quenching constant and $\tau_0$ is the lifetime at zero oxygen. The Stern-Volmer equation is valid for a homogenous oxygen distribution and after excitation with a pulse of light of which the lifetime is much shorter than $\tau$. In case of a non-homogenous oxygen distribution inside the measurement volume, a reliable estimation of the average PO$_2$ can be made by the rectangular distribution method (RDM) [26, 27]. A schematic presentation of PpIX-TLST is provided in Fig 1.

The signal/noise ratio (SNR) of resulting traces was calculated and defined as the ratio of maximum signal amplitude to the peak-to-peak noise. COMET$^®$ evaluates signal quality, which is calculated from the SNR value; the relationship between signal quality and SNR is non-linear and an increase of one in SNR is approximately 1% in signal quality up till a SNR of 50 [10].

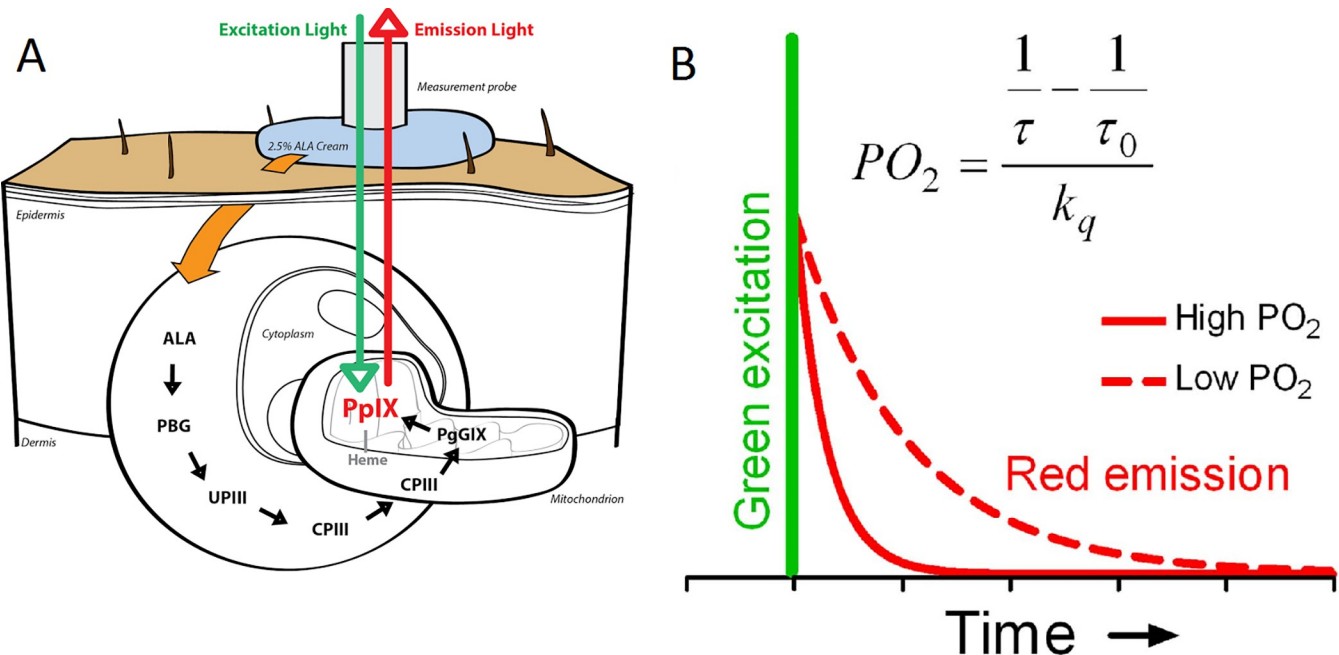

**Fig 1.** (a) Principle of protoporphyrin IX-Triplet State Lifetime Technique. The pathway by which topical ALA administration enhances mitochondrial PpIX levels and the principle of delayed fluorescence detection after an excitation pulse with green (510 nm) light. Emission light is the delayed fluorescence (red light, 630–700 nm) and its lifetime is oxygen-dependent. (b) PpIX emits delayed fluorescence after excitation by a pulse of green (510 nm) light. The delayed fluorescence lifetime is oxygen-dependent according to the Stern–Volmer equation (inset), in which kq is the quenching constant and τ0 is the lifetime at zero oxygen. ALA, 5-aminolevulinic acid; CPIII, coporporphyrinogen III; PBG, porphobilinogen; PO2, oxygen tension; PpIX, protoporphyrin IX; UPIII, urporphyrinogen II. Reproduced from the original publication in Critical Care with permission from Harms et al. [28].

## Procedures (measurement)

Induction of anesthesia was performed by the attending anesthesiologist. Intraoperative monitoring, ECG-based heart rate, invasive blood pressure, body temperature, skin temperature and oxygen saturation, inspired oxygen fraction ($FiO_2$), anesthesia infusion rate and vasopressor pump setting were part of the standard monitoring and were stored in the electronic patient data management system. All patients were kept normothermic with use of a warm air blanket. Normovolemia was pursued with intravenous crystalloids based on the pulse pressure variation index (threshold above 13).

MitoPO$_2$ measurements were performed through the use of the COMET® monitor (Photonics Healthcare, Utrecht, the Netherlands). In order to upregulate PpIX, a self-adhesive patch containing 8mg 5-aminolevulinc acid (ALA) (Alacare®, photonamic GmbH und Co. KG, Pinneberg, Germany) was applied on the sternal skin. Alacare ® is used off-label for mitochondrial oxygen tension measurements. To enhance ALA penetration adequate skin preparation proved essential. Hair was shaved (if present) and the skin was rubbed with a fine abrasive pad to remove the top parts of the stratum corneum. During ALA application, the skin was protected from light. Application of at least 5 hours allowed for a suitable concentration of PpIX to be synthesized in order to enable measurements of mitoPO$_2$. For the logistical reason that the surgeries often started at 8 a.m. and it was not considered patient-friendly to place the plaster in the middle of the night, longer application times were accepted. After the induction of anesthesia, the ALA patch was removed and the measuring probe applied to the ALA-treated skin. The mitoPO$_2$ was automatically measured every 5 minutes during the operation.

In addition to the mitoPO$_2$, tissue oxygenation saturation and perfusion parameters were measured intraoperatively using the O2C (oxygen to see version 2424, Lea Medizintechnik GmbH, Giessen, Germany). The O2C measures three parameters: The local capillary venous saturation (StO$_2$), the local velocity of blood given in velocity units (VU) and the local microvascular blood flow given in flow units (AU). Both the COMET® Skin Sensor and the O2C probe (LFX-43) were positioned on the sternum next to each other.

All measurements were performed from the start of surgery until the end of surgery, in order to exclude the effects of induction of anesthesia and the accompanying influences of medication and pre-oxygenation on mitochondrial and microvascular parameters. To observe the feasibility and stability of the mitoPO$_2$ measurements the outcome measures included: mitoPO$_2$ (mmHg), local capillary venous saturation (%), flow (AU), inspired oxygen fraction (FiO$_2$, %), peripheral oxygen saturation (SpO$_2$, %) and skin temperature (degrees Celsius).

### Statistical analysis

A sample size calculation was performed using G*Power software and was based on previous research in healthy volunteers [14, 29]. Because of technical improvements less device-induced variation was expected and therefore a lower standard deviation was assumed. MitoPO$_2$ was suspected to be slightly higher due to the continuous oxygen supplementation during surgery. For a paired t-test, a sample size of 20 patients was calculated with an assumed mean difference of 12 mmHg and standard deviation of 18 mmHg, a type I error probability of 0.05 and a power of 80%.

Statistical analyses were performed using Graphpad Prism 8, IBM Statistics SPSS 26 and R Statistics [30–32] As the O2C measures several times a minute the data was averaged over 60 seconds to allow fluctuation in the minutes range. Descriptive statistics were used to describe demographic parameters. Continuous variables are described as mean and standard deviation (SD). Normality was assessed by visualizing the data using Q-Q plots, histograms and the residuals.

A paired t-test was used to compare the mean baseline mitoPO$_2$ value to the mean 'end-of-surgery' mitoPO$_2$ value. Furthermore, a mixed linear mixed effects model (LMM) was created to analyze the association between mitoPO$_2$, skin temperature, microvascular blood flow, time point, StO$_2$, MAP, heart rate, SpO$_2$ and FiO$_2$. In this model, the above-mentioned covariates were entered as fixed effects and subject number was entered as random effect. It should be noted that measurement errors were not corrected for. For this analysis, the lme4 package in R was used.

### Results

A total of 24 neurosurgery patients were included in the study, 4 patients were excluded from further analysis. One patient was excluded due to measurement failure, one patient did not apply the ALA patch before surgery, the operation was rescheduled for another patient, and the positioning during neuro surgery changed from supine to prone position for the last patient thus interfering with our sternal measurements. For 2 of the remaining 20 patients the O2C data is missing because the O2C monitor was unavailable. Surgical and patient characteristics are summarized in Table 1. On average, the ALA patch was applied 17 ± 3.3 hours before surgery. In all patients, sufficient signal quality for mitochondrial oxygenation measurements was reached. Mean signal quality at the beginning of the measurements was 47 ± 21% and at the end 45 ± 19%. It was observed that signal quality was greatly affected by background light. In the OR, this was predominantly caused by the surgical lighting and could be solved by adequate shielding of the probe against light.

**Table 1. Baseline patient characteristics and clinical data.**

| Baseline patient characteristics and clinical data | |
|---|---|
| **Variables** | **Total (n = 20)** |
| Age (yr)[a] | 56.2 (± 11.7) |
| Sex | |
| Male/female | 10/10 |
| BMI[a] | 26.8 (± 3.7) |
| Medical history | |
| Chronic obstructive pulmonary disease | 10% (2) |
| Hypertension | 15% (3) |
| Diabetes Mellitus | 5% (1) |
| Vascular event | 22% (4) |
| Surgery type | |
| Resection of a vestibular schwannoma | 10% (2) |
| Hypophysectomy | 30% (6) |
| Trepanation | 60% (12) |
| ASA I/II/III/IV | 4/8/7/1 |
| Blood pressure[a] (mmHg) | |
| Systolic | 134.4 (±15.2) |
| Diastolic | 80.7 (±10.5) |
| Heartrate (beats/min)[a] | 74.8 (± 15.4) |
| Blood loss (ml) [a] | 217 (± 202) |

[a] = mean (standard deviation), n = number of included patients, yr = years

Abbreviations: ASA; American Society of Anesthesiologists, BMI; body mass index

Anesthetic and analgesic agents were selected by the anesthesiologist who performed the anesthesia. Induction of anesthesia was performed using an induction bolus of propofol in 19 patients (mean dose 179 mg ± 41) or thiopental in 1 patient (500 mg) in combination with sufentanil (7 patients, mean dose 22 mcg ± 4), fentanyl (3 patients, 500 mcg) or piritramide (1 patient, 25 mcg). For maintenance of the anesthesia, all patients received propofol (mean dose 7 mg/kg/hour). In 5 patients this was combined with inhalation of sevoflurane (mean end tidal concentration 1.4% ± 0.6) and in 3 patients with isoflurane (mean end tidal concentration 0.9% ± 0.2). Maintenance of analgesia was achieved using remifentanil in 19 patients (mean dose 11 mcg/kg/hour ± 2) and with sufentanil in 1 patient (26 mcg/hour).

## Hemodynamic variables

The surgical conditions of all 20 patients were hemodynamically stable (Fig 2). However, low doses of vasopressors were required in order to maintain MAP above 60 mmHg for all patients. 14 patients had noradrenaline administered (0.05 mcg/kg/min ± 0.02 mcg/kg/min) and phenylephrine was used in 6 other patients (0.28 mcg/kg/min ± 0.14 mcg/kg/min). No rises in serum lactate level were observed. Local capillary venous saturation and microvascular blood flow were stable during surgery as presented in Fig 3B and 3C. The mean skin temperature increased from baseline (start surgery) over time (Fig 3D). In the first hour of operation, the mean skin temperature increased more than 1˚C.

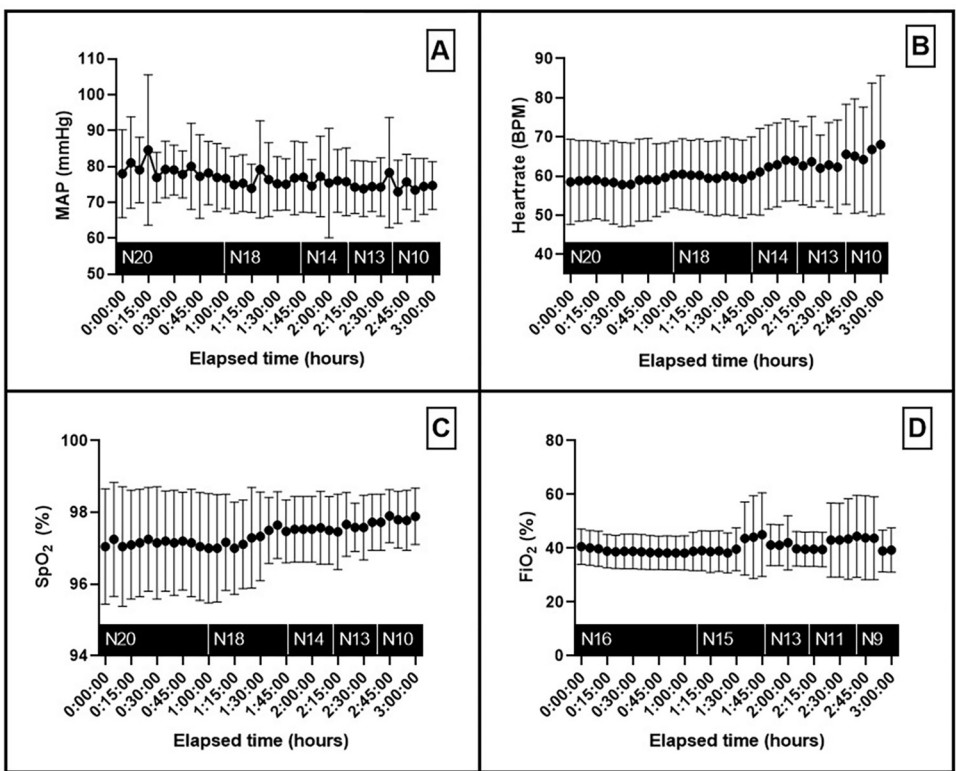

**Fig 2. Overview of the hemodynamics parameters of the included patients during neurosurgery.** A. Mean Arterial Pressure B. Heart rate. C. Peripheral oxygen saturation. D. Fraction of inspired oxygen. Data presented as mean (dot) and ± standard deviation. N; number of patients, MAP; mean arterial pressure, SpO2; peripheral oxygen saturation, FiO$_2$; fraction of inspired oxygen.

## MitoPO$_2$

Mean baseline mitoPO$_2$ (± SD) was 60 ± 19 mmHg, and during surgery the mean mitoPO$_2$ remained between 40 and 60 mmHg. However, a slow decline from baseline was observed during the first hour of surgery (Fig 3A). The coefficient of variation of mitoPO$_2$ was 0.31. When compared to the 'end of surgery' mitoPO$_2$ ($M = 47$, $SD = 14$) using a paired t-test, mean baseline mitoPO$_2$ ($M = 60$, $SD = 19$) was significantly higher ($t(19) = 3.845$, $p = 0.001$).

### Regression analysis using LMM

Two LMM models were created, the first model only included skin temperature and time point as fixed effects and subject number and time point as random effects, whilst the second model included skin temperature, flow, StO$_2$, MAP, HR, SpO$_2$ and FiO$_2$. Both models indicated a significant association between skin temperature and mitoPO$_2$ ($p < 0.001$). With every increase of 1 degrees Celsius in skin temperature, mitoPO$_2$ decreased by 1.76 mmHg in the first model and by 2.85 mmHg in the second model. Additionally, the time point of the measurement, microvascular blood flow, StO$_2$ and SpO$_2$ were correlated with mitoPO$_2$ ($p < 0.001$, $p = 0.015$, $p = 0.017$ and $p < 0.001$). The marginal R$^2$ of the first model was 3.4% and the conditional R$^2$ 84.2%. In the second model, which included all the covariates, the marginal R$^2$ is 20.8% and the conditional R$^2$ 87.8%. Further detail of the models can be found in S1 and S2 Tables in the S2 File.

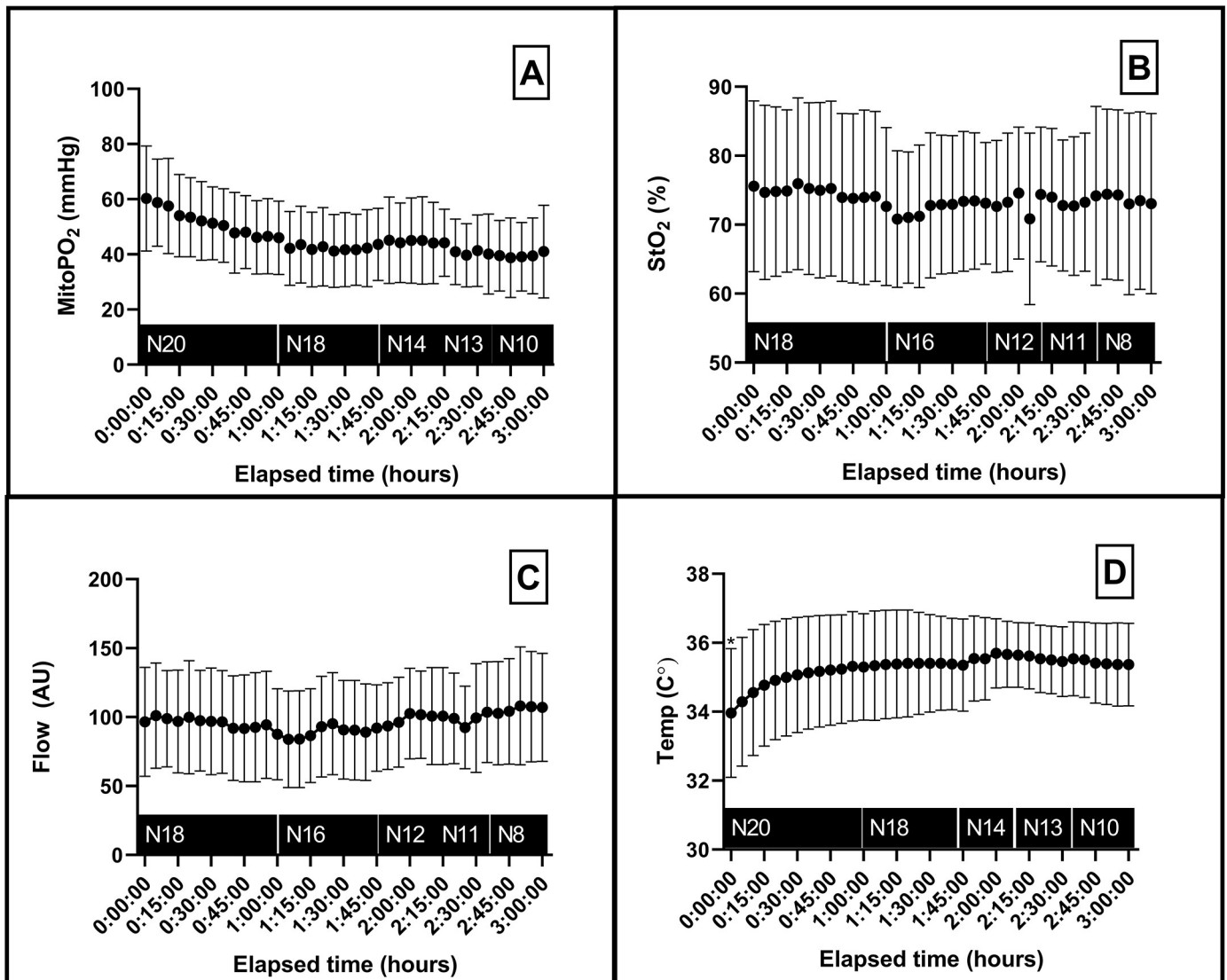

**Fig 3. Overview of the microcirculatory parameters of the included patients during neurosurgery.** A. Mitochondrial oxygen tension as measured by the COMET. B. Local capillary venous saturation as measured by the O2C. C. Mean flow in microcirculation measured with the O2C. D. Mean skin temperature. Data presented as mean (dot) and ± standard deviation. N; number of patients, MitoPO2; mitochondrial oxygen tension, AU; arbitrary units, Temp; skin temperature.

In Fig 4. A single case of the neurosurgery cohort is presented. This case is notable because of long operation time. A 35-year-old woman who had surgery on a difficult-to-reach vestibular schwannoma had a surgery time of approximately six hours. She did remain hemodynamically stable with a mean map of 71 ± 8 mmHg. A total of 600 ml blood was lost, which was supplemented with crystalloid and colloid solutions to maintain a normovolemic fluid balance. This single case highlights the possibility to have stable mitoPO$_2$ measurements during long periods of time under stable hemodynamic conditions.

## Discussion

This study shows that it is feasible to measure the mitoPO$_2$ with the COMET® during surgery in real-world situations. It is important to note that covering the measurement surface and

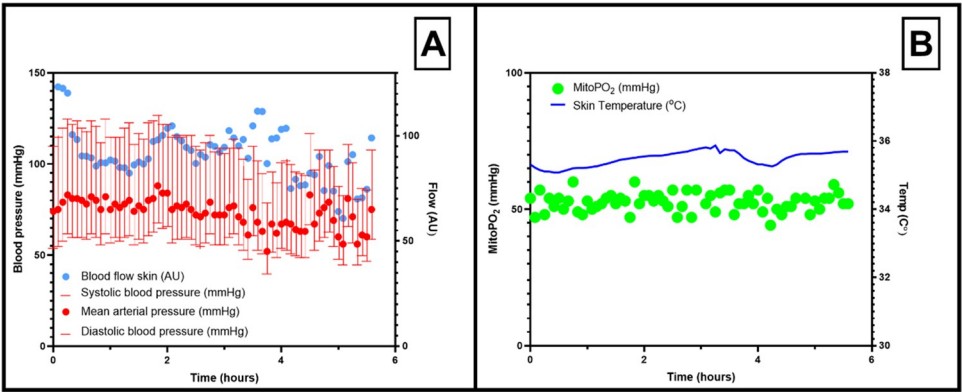

**Fig 4. Representative case of stable mitoPO$_2$ measurements during an operation of more than six hours.** MitoPO$_2$; mitochondrial oxygen tension, AU; arbitrary units, Temp; skin temperature.

protecting it against ambient light is essential for getting an adequate mitoPO$_2$ measurement. Our study shows that mitoPO$_2$ is a stable parameter in hemodynamically uneventful intraoperative circumstances.

In hemodynamically stable patients a mean mitoPO$_2$ between 40 and 60 mmHg was found and the baseline mitoPO$_2$ was $60 \pm 17$ mmHg. Previously published results in healthy volunteers concluded a mean mitoPO$_2$ value of $44 \pm 17$ mmHg [14]. Higher mitoPO$_2$ values were expected in this study, as a higher inspiratory oxygen fraction was used during surgery leading to a higher arterial partial pressure of oxygen and thus a higher tissue partial pressure of oxygen. Since mitochondrial oxygen tension reflects the balance between oxygen supply and oxygen demand [33, 34], a decreased metabolism during general anesthesia could be a contributing factor to the higher mitoPO$_2$ witnessed in this cohort [35]. A decreased mitochondrial metabolic rate has not only been established during general anesthesia, but also with various lifestyle-related diseases such as, neurodegenerative disease [36], cardio-vascular disease [37] and obesity [38, 39]. Septic patients are another group of patients whom exhibit a reduced oxygen metabolism and mitochondrial failure [40]. Recently a study in patients with severe sepsis showed a relatively high mitoPO$_2$ ($61 \pm 10$ mmHg) [22].

Despite the initial intention to measure mitoPO$_2$ in the operation theatre under stable conditions, it was not possible to ensure that all parameters remained stable. Skin temperature, timing of the measurement, microvascular blood flow, StO$_2$ and SpO$_2$ proved to be associated with mitoPO$_2$. The estimates of microvascular blood flow and StO$_2$ were rather small (0.11 increase in mitoPO$_2$ for 1 unit increase in microvascular blood flow and 0.18 increase in mitoPO$_2$ for 1 unit increase in StO$_2$), and therefore do not seem clinically relevant. Furthermore, when visualized (Fig 3), only mean skin temperature showed changes at the same time as mitoPO$_2$. This implies that skin temperature is associated with the mitoPO$_2$. However, the marginal $R^2$ in the model for skin temperature was small, whilst the conditional $R^2$ was large, which insinuates that a large part of the variation in mitoPO$_2$ is explained by the random effect part. Therefore, no definite conclusions on the association between mitoPO$_2$ and skin temperature can be drawn from this study. The association should be investigated in a specifically designed trial.

Theoretically, the rise in skin temperature and subsequent decrease in mitoPO$_2$ can be explained by an increased cellular metabolism as result of the higher temperature. In turn causing an increased oxygen demand and thus decreased mitoPO$_2$ values. This higher rate of energy consumption cause cells to rapidly break ATP down to adenosine diphosphate (ADP)

and an inorganic phosphate group (Pi). In this process adenosine monophosphate (AMP) and [$H^+$] are released which are both strong vasodilators resulting in increased blood flow and tissue oxygen delivery [41, 42]. The data found by this study supports the afore mentioned, as a rise in temperature gives a drop in intracellular $PO_2$ (mitoPO$_2$) whilst the cutaneous microvascular blood flow and StO$_2$ did not decrease.

Ubbink et al. [10] have previously presented a case in which mitoPO$_2$ decreased as a result of less microvascular blood flow because of the administration of a bolus of clonidine (an α2-adrenergic agonist resulting in vasodilation). However, no changes in the StO$_2$ were measured. An alternative explanation of the differences between mitoPO$_2$, StO$_2$ and flow could be the different wavelengths used by the different measurement techniques. StO$_2$ is measured with the absorption spectrum of visible light, mitoPO$_2$ is measured with green light (515 nm) and velocity and flow are measured with based on the Doppler shift of the detected laser light caused by the erythrocytes (LEA site). All these different wavelengths give different tissue penetrations [43] and thus measure in different tissue compartments and could therefore show different results. These findings suggest that the ability to measure oxygen availability directly at the cellular level through mitoPO$_2$ measurements provides complementary data on the microcirculation and can enable new insights into patient management.

## Clinical interpretation and value of mitoPO$_2$ measurements

In current clinical practice hemodynamic optimization during major surgery remains challenging, partly due to the absence of an adequate end-point for tissue hypoxia. Consequently, patients are often over or under resuscitated.

This study showed a stable mitoPO$_2$ with values between 40 and 60 mmHg under stable operative circumstances. A recently published case report described the mitoPO$_2$ during acute perioperative blood loss [25]. A striking observation in this case report was that primarily the mitoPO$_2$ decreased whilst the standard parameters, such as blood pressure, heart rate and pulse variation index did not. Hemoglobin levels and SpHb were reduced but not to values indicating a direct need for a blood transfusion. Meanwhile the serum lactate levels also remained low. The standard parameters therefore, supported the physician's decision not to give a blood transfusion despite the large amount of blood loss. However, in a later phase, after continuous blood loss, heart rate and capillary blood flow did show a response to the bleeding. As a consequence of the changed parameters, the patient was resuscitated with red blood cells. After transfusion a rapid increase of mitoPO$_2$ was observed, from values below 10 mmHg to up to 40 mmHg. This example illustrates the potential of mitoPO$_2$ to respond to hemodynamically unstable situations. Furthermore, it demonstrates that during hemodynamically unstable conditions mitoPO$_2$ values are far below the normal standard deviation, as values of less than 10 mmHg were measured. To further illustrate the behavior of mitoPO$_2$ in hemodynamically unstable conditions, two other cases can be found in the S1 File.

The aforementioned paragraph highlights the potential of mitoPO$_2$ as a parameter for early detection of physiologic decompensation, which seems more sensitive to changes compared to other parameters. This has previously also been demonstrated in a porcine model in which the pigs underwent hemodilution and as a result the mitoPO$_2$ decreased prior to StO$_2$, lactate and a decrease of the MAP [44]. Moreover, the epidermal mitoPO$_2$ could also be utilized as a tissue hypoxia indicator for the tissue oxygenation of different organs. During sympathetic activation, blood flow is directed away from the skin to the vital organs [45]. Therefore, it is one of the first organs that becomes hypoxic during hemodynamically unstable conditions [46, 47].

Since many factors are involved in maintaining an adequate cellular oxygenation, we suggest that perioperative hemodynamic monitoring, blood transfusion and fluid therapy

strategies during complex surgeries should not be based on standard care intraoperative parameters alone. During these circumstances the addition of mitochondrial oxygenation and microvascular flow and saturation measurements can be essential to prevent cellular hypoxia and organ damage in order to improve long-term outcome [13, 14, 21]. The added value of the mitoPO$_2$ measurements during major surgeries must be further examined in future studies. The present study is an important step towards monitoring of mitoPO$_2$ in the operating theatre.

In this study the sternum was chosen as our measurement site for the mitoPO$_2$ as it is a centrally located on the human body. We hypothesize that the more distal the measurement site is, the more difficult it becomes to interpret changes in mitoPO$_2$. This can be due to a myriad of factors such as vasoconstriction, temperature changes, and altered epidermal metabolism. However, mitoPO$_2$ can be measured on other sites as well, as a recently published study in which the mitoPO$_2$ of cardiothoracic surgery patients was measured on the upper arm [48]. The reason for this was that the probe could not be placed on the sternum due to logistical constraints. The upper arm measurement site seems to be a good alternative as its location is still relatively central.

Mean application time of the Alacare$^®$ plaster was 17 hours. For the intended use of Alacare$^®$, namely in combination with photodynamic therapy for the treatment of actinic keratosis, the recommended application time is four hours [49]. However, in a clinical setting exact timing appeared to be an issue and we have gained experience with longer application times. Up until now we have been able to obtain adequate measurements with application times ranging from 5 to 22 hours.

As described by Ubbink et al. [10], the COMET$^®$ monitor displays a signal quality percentage taking into account the SNR of the delayed fluorescence signal. However, the displayed signal quality parameter has a non-linear relationship with SNR and appears to be affected by other factors than solely the SNR. As this is the first experience with the COMET$^®$ monitor in a clinical setting, we have subsequently analyzed the raw data of the delayed fluorescence signal. Based on this analysis, we can conclude that none of the measurements have had a SNR which is considered to cause unreliable measurements (i.e. SNR < 20%, [10]), whilst the displayed signal quality could be as low as 9%. It appears that for the clinical environment the current implementation of signal quality calculation in the COMET device is quite conservative. Therefore, the authors suggest further evaluation of this parameter and future optimization of signal quality feedback to the end-user in the clinical setting.

## Study limitations and strengths

The presented study is the first study to demonstrate mitoPO$_2$ measurements under hemodynamically stable intraoperative conditions. Despite the intention however, the temperature of the skin did not remain stable, whereby skin temperature was associated with mitoPO$_2$, as demonstrated by the LMMs.

Our choice for an observational study design introduces both strength and limitations. The choice for hemodynamically stable neurosurgical patients as study population enabled us to explore possible confounders of the measurement beside hemodynamic alterations. However, hemodynamic unstable conditions are of interest, as we expect the mitoPO$_2$ to react to the changes. Furthermore, we did not include a control group, nor was it possible to analyze possible interventions to increase mitoPO$_2$. Both will have to be examined in future trials.

## Conclusions

To conclude, this study shows the feasibility and applicability of measuring mitoPO$_2$ in the operating theatre using the COMET$^®$ monitor. Moreover, the mitoPO$_2$ measurements remain

feasible even during prolonged surgery. Most importantly, the parameter mitoPO$_2$ is stable under hemodynamically stable conditions while otherwise being sensitive to physiological alterations.

## Supporting information

**S1 File.**
(DOCX)

**S2 File.**
(DOCX)

## Author Contributions

**Conceptualization:** Floor A. Harms, Luuk H. Römers, Rineke Janse.

**Data curation:** Floor A. Harms.

**Formal analysis:** Floor A. Harms, Mark A. Wefers Bettink.

**Investigation:** Lucia W. J. M. Streng, Mark A. Wefers Bettink, Luuk H. Römers, Rineke Janse.

**Methodology:** Floor A. Harms.

**Project administration:** Floor A. Harms, Lucia W. J. M. Streng, Rineke Janse.

**Supervision:** Floor A. Harms, Robert J. Stolker, Egbert G. Mik.

**Visualization:** Lucia W. J. M. Streng, Calvin J. de Wijs.

**Writing – original draft:** Floor A. Harms, Lucia W. J. M. Streng.

**Writing – review & editing:** Lucia W. J. M. Streng, Calvin J. de Wijs, Luuk H. Römers, Rineke Janse, Robert J. Stolker, Egbert G. Mik.

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
