## [Editor Report · Decision Letter 0]

1 Nov 2021

PONE-D-21-32161Monitoring of mitochondrial oxygen tension in the operating theatre: an observational study with the novel COMET® monitor.PLOS ONE

Dear Dr. Harms,

Thank you for submitting your manuscript to PLOS ONE. After careful consideration, we feel that it has merit but does not fully meet PLOS ONE’s publication criteria as it currently stands. Therefore, we invite you to submit a revised version of the manuscript before the peer reviewing process.

We look forward to receiving your revised manuscript.

Kind regards,

Yu-Chang Yeh, M.D., Ph.D.

Academic Editor

PLOS ONE

Journal Requirements:

2. We note that you have a patent relating to material pertinent to this article. Please provide an amended statement of Competing Interests to declare this patent (with details including name and number), along with any other relevant declarations relating to employment, consultancy, patents, products in development or modified products etc. Please confirm that this does not alter your adherence to all PLOS ONE policies on sharing data and materials, as detailed online in our guide for authors http://journals.plos.org/plosone/s/competing-interests by including the following statement: "This does not alter our adherence to  PLOS ONE policies on sharing data and materials.” If there are restrictions on sharing of data and/or materials, please state these. Please note that we cannot proceed with consideration of your article until this information has been declared.

Additional Editor Comments:

It is impressive that the authors try to investigate the feasibility and applicability of the COMET measurements in the operation theatre during stable hemodynamic conditions. Please consider revising the manuscript to facilitate the peer reviewing process.

1. Consider deleting the two cases and the related contents.

2. More information of intraoperative variables will be helpful, e.g., maintenance of anesthesia, amount of fluid supplement, and heart rate.

3. LINE 104, the equation requires revision.

4. Please clarify the clinical interpretation of mitoPO2.

5. Please clarify the “great value” for improving hemodynamic management, e.g., poor mitoPO2 masked by normal hemodynamic management under specific condition.

6. Please list the limitations and strengths.

---

## [Author Response · Author response to Decision Letter 0]

17 Jan 2022

Dear Editors, 

Thank you for your time and expertise reviewing our manuscript. We have revised the manuscript based on the given comments, in detail: 

1. Consider deleting the two cases and the related contents. Author’s reply: we have deleted the cases and the related contents

2. More information of intraoperative variables will be helpful, e.g., maintenance of anesthesia, amount of fluid supplement, and heart rate. Author’s reply: we have added extra information of intraoperative variables, in particular maintenance of anesthesia, heart rate, oxygen saturation and inspired oxygen fraction

3. LINE 104, the equation requires revision. Author’s reply: We have revised the equation in the manuscript.

4. Please clarify the clinical interpretation of mitoPO2. Author’s reply: Since many factors are involved in maintaining an adequate cellular oxygenation, we suggest that perioperative hemodynamic monitoring, blood transfusion and fluid therapy strategies during complex surgeries should not be based on standard care intraoperative parameters alone. During these circumstances the addition of mitochondrial oxygenation and microvascular flow and saturation measurements can be essential to prevent cellular hypoxia and organ damage in order to improve long-term outcome. We have added this to the discussion section.

5. Please clarify the “great value” for improving hemodynamic management, e.g., poor mitoPO2 masked by normal hemodynamic management under specific condition. Author’s reply: In the current clinical practice hemodynamic optimization during major surgery remains challenging remains, partly due to the absence of an adequate end-point for tissue hypoxia. Therefore patients are often over or under resuscitated. Mitochondrial oxygenation and microvascular flow and saturation measurements can be essential to gain insight into the microcirculation and prevent cellular hypoxia and organ damage in order to improve long-term outcome. We have added this to the discussion section.

6. Please list the limitations and strengths. We have added the limitations and strengths 

of the study to the discussion section.

In addition to the revision of the manuscript we would like to add C.J. de Wijs to the authors list due to his contributions in reviewing and revising the manuscript and change the first authorship of F.A. Harms to co-first authorship with L.W.J.M. Streng considering her contributions to the measurements, data processing, writing and revising of the manuscript. All authors have given consent for this change.

Kind regards, 

Floor Harms

---

## [Decision Letter · Decision Letter 1]

22 Feb 2022

PONE-D-21-32161R1Monitoring of mitochondrial oxygen tension in the operating theatre: an observational study with the novel COMET® monitor.PLOS ONE

Dear Dr. Harms,

Thank you for submitting your manuscript to PLOS ONE. After careful consideration, we feel that it has merit but does not fully meet PLOS ONE’s publication criteria as it currently stands. Therefore, we invite you to submit a revised version of the manuscript that addresses the points raised during the review process.

We look forward to receiving your revised manuscript.

Kind regards,

Yu-Chang Yeh, M.D., Ph.D.

Academic Editor

PLOS ONE

Reviewers' comments:

Reviewer's Responses to Questions

**Comments to the Author**

1. If the authors have adequately addressed your comments raised in a previous round of review and you feel that this manuscript is now acceptable for publication, you may indicate that here to bypass the “Comments to the Author” section, enter your conflict of interest statement in the “Confidential to Editor” section, and submit your "Accept" recommendation.

Reviewer #1: (No Response)

Reviewer #2: (No Response)

Reviewer #3: (No Response)

2. Is the manuscript technically sound, and do the data support the conclusions?

Reviewer #1: Yes

Reviewer #2: Yes

Reviewer #3: No

3. Has the statistical analysis been performed appropriately and rigorously? 

Reviewer #1: N/A

Reviewer #2: Yes

Reviewer #3: Yes

4. Have the authors made all data underlying the findings in their manuscript fully available?

Reviewer #1: Yes

Reviewer #2: Yes

Reviewer #3: Yes

5. Is the manuscript presented in an intelligible fashion and written in standard English?

Reviewer #1: Yes

Reviewer #2: Yes

Reviewer #3: Yes

6. Review Comments to the Author

Reviewer #1: In their paper, "Monitoring of mitochondrial oxygen tension in the operating theatre: an observational study with the novel COMET® monitor," the authors demonstrated the use of the novel COMET® monitor. They illustrate real-time measurements of mitoPO2 in two different patients in hemodynamically unstable situations. The manuscript presents case reports that confirm the validation of the proposed techniques. It demonstrates the reliable and significant applicability of this technique in routine practise.

Comments:

1.I would have expected more information in the introduction about mitoPO2 and the importance of its measurement. Comparison with other available methods and research of other groups in this field. A considerable amount of self-citation appears in the manuscript.

2. page6 line 104: equation is not well formatted.

3. page7 line 134: The use of ALA was indicated at least 5 hours before surgery. However, later in the text (page 10 line 176) the authors mention 17 hours before surgery. It would be useful to specify a time period when the measurement should and could be done, since ALA is metabolised later in the tissue.

4) page 15 line 281: The authors mention previously published results, but there is no reference to this work.

5) The limitations and advantages of the experimental design should also be discussed.

6) page 16 line 312: mitoPO2 (lower index)

Reviewer #2: The study aims to assess the feasibility and applicability of the COMET monitor in 20 patients undergoing neurosurgery under stable environmental conditions. Based on the PpIX-TSLT the COMET allows non-invasive, in-vivo measurements of mitochondrial oxygenation (mitoPO2). Next to mitoPO2 measurements, tissue oxygenation and local blood flow (Oxygen to see, O2C) and several other covariates were documented. In hemodynamically stable patients, the mitoPO2 remained relatively stable. Nonetheless, especially in the first hour of surgery mitoPO2 tended to decrease in association with increasing probe temperature of the COMET.

Please find the detailled review in the attached pdf-file.

Reviewer #3: This reviewer reviews this manuscript for the first time and is not aware of the original manuscript.

The authors studied mitochondrial oxygen tension (mitoPO2) in general surgical patients using a new non-invasive device. The goal of this study is stated as testing the feasibility of the device for intraoperative use. The authors also hypothesized that the mitoPO2 would remain stable during the hemodynamically stable conditions. The authors successfully used the device intraoperatively and concluded the mitoPO2 measured by this device is reliable. The conclusion is not supported due to lack of the validity of this mitoPO2 monitoring. The authors’ hypothesis is also not supported due to the observed decline of mitoPO2. Overall, the study has major issues and the clinical and scientific contribution of this study is unclear.

Major Comments:

There is no discussion of the validity of the mitoPO2 monitor. What the face validity of this monitoring device is, in other words, if this device measures what it claims to measure, mitochondrial oxygenation, is not discussed. This discussion needs to include preclinical studies that are foundational for the development of this monitor. For example, are there animal studies performed to show the validity of this measurement? Was this measurement compared to other types of the mitoPO2 measurement or any existing related monitors? Content validity, whether this measurement covers all of the thing the authors are trying to measure, may deserve a consideration, too. If the device has not been tested for the validity, feasibility testing seems to be useless.

A major scientific question is the role of mitoPO2. A possible role of mitoPO2 may be an early detection of tissue/cellular hypoxia before currently existing hemodynamic and oxygen monitors can do so. If that is the question, I believe that large animal studies, such as pigs, can answer the question in much more controlled conditions. Are these studies done in animals?

Related to the above question, characteristics of the measurement needs to be explained. What is the temporal resolution of this measurement? Can the measurement reflect the mitoPO2 change at the second, minute or longer-level? How is the 5-min interval determined? The temporal resolution is important for interpreting the data and the usefulness as a monitor. In addition, how does the skin mitoPO2 reflect major organ mitoPO2? Are they correlated? If there are discrepancies between skin and other organs, the monitoring of the skin mitoPO2 may not be helpful. Further, the authors discuss that the decline of mitoPO2 during surgery may be associated with the skin temperature change. How the skin and ambient temperatures and warming methods affect the mitoPO2 should be easily tested on animals.

This study measured mitoPo2 in the limited time period (stable intraoperative) and in the limited patients. Anesthesia induction and emergence, hemodynamic instability, blood transfusion, moderate – high dose vasopressors are all excluded. Patients with presence of mitochondrial diseases, pregnancy or lactation are also excluded. However, the authors discuss a case report in which the mitoPO2 measurement was useful in a patient who had a large amount of blood loss. Therefore, it is not clear to this reviewer why the authors excluded all the instabilities, which seems to be a missed opportunity.

Other Comments:

Are there any possible adverse effects of Alacare? Is Alacare absorbed systemically? Is Alacare approved for clinical use in the European nations?

Line100: “Hypertension was medically induced during the hemostasis phase”. Is this induced hypertension for neurosurgical procedures?

Line 171: What are the causes of low signal quality?

Line 198: When exactly the baseline mitoPO2 is measured? Is this before or after surgery started. Why is the awake mitoPO2 not included?

Line 276: The effect of anesthesia should be tested in preclinical animal models.

Figure 1 and 2: What are N20, N16, etc.? The averaged data across patients may not be helpful as all the surgical stimulation cannot be controlled. Are the data normalized or not normalized?

Diagram for the mitoPO2 measurement mechanism may be helpful for the readers.

7. PLOS authors have the option to publish the peer review history of their article (what does this mean?). If published, this will include your full peer review and any attached files.

Reviewer #1: No

Reviewer #2: No

Reviewer #3: No

---

## [Author Response · Author response to Decision Letter 1]

5 Aug 2022

Answers to the comments of the reviewers on manuscript: Monitoring of mitochondrial oxygen tension in the operating theatre: an observational study with the novel COMET® monitor.

Reviewer #1: 

Comments:

1 .I would have expected more information in the introduction about mitoPO2 and the importance of its measurement. Comparison with other available methods and research of other groups in this field. A considerable amount of self-citation appears in the manuscript. 

Answer: We have taken note of the reviewers suggestion and added an extra paragraph in the introduction on available different methods of measuring tissue PO2 and their limitations. 

2. page6 line 104: equation is not well formatted.

Answer: The equation has been re-formatted. 

3. page7 line 134: The use of ALA was indicated at least 5 hours before surgery. However, later in the text (page 10 line 176) the authors mention 17 hours before surgery. It would be useful to specify a time period when the measurement should and could be done, since ALA is metabolised later in the tissue.

Answer: This is a very useful suggestion. We have started our experiments with an application time of four hours, which is sufficient in optimal measuring circumstances. However, in clinical studies exact timing appeared to be an issue (e.g. due to delayed procedures and rescheduling) and we gained experience with longer application times. Additionally, longer application times appeared more convenient if we want to measure during surgeries, because four hours before the start of the surgery would mean application in the middle of the night in early surgeries. Up until now we have been able to obtain adequate measurements with application times ranging from 4 to 21 hours. However, based on experience we now do recommend at least 5 hours of application time when the measurements are performed in a clinical setting because of the better signal quality compared to 4 hours.

4) page 15 line 281: The authors mention previously published results, but there is no reference to this work.

Answer: We have added a reference after the above mentioned line. 

5) The limitations and advantages of the experimental design should also be discussed.

Answer: The limitations and advantages have been added to the discussion. 

6) page 16 line 312: mitoPO2 (lower index)

Answer: The 2 has been changed to subscript.

Reviewer #2: 

Abstract 

Minor:L35: 60 ± 17 does ± refer to SD? Please clarify 

Answer: 60 ± 17 does refer to mean and standard deviation. We have also added this to the paragraph. 

L36: Based on Figure 2 skin/probe temperature rose > consider adding with “increasing“ temperature 

Answer: The word increasing has been added to the paragraph.

L38: stable and unstable: in the current version of the manuscript the authors only present data on hemodynamically stable patients 

Answer: The word unstable has been removed from line 38. 

L40: Can the authors please comment on the exact meaning of “reliable”.

Answer: The sentence has been rephrased. 

L40-41: Please consider rephrasing the sentence. 

Answer: The sentence has been rephrased

L41: Basically it was in neurosurgical patients. 

Answer: ‘Major surgery’ has been changed to ‘measurements in the perioperative trajectory’.

Introduction 

Major: 

L74: „longitudinal observational pilot study“. I think this is not a classical longitudinal study. 

Answer: All mentioning of a longitudinal study has been changed to “observational pilot study”.

Minor: 

IL61 […] derived from muscle biopsies > consider adding a reference 

Answer: The paragraph has been rephrased. 

Methods 

The methods part including trial design, setting and eligibility criteria as well as the principle of mitoPO2 measurements is well written. 

Major: 

L163-169: The sample size planning is not clear: “A sample size of 20 patients was calculated with an assumed mean difference of 12 mmHg and standard deviation of 18 mmHg, a type I error probability of 0.05 and a power of 80%.“ For what kind of test? In addition, there are no tests reported within the manuscript. Please clarify.

Answer: The sample size has been calculated for a t-test using G*power. We have added this in the paragraph. 

Minor: 

L175: one bracket too much

Answer: The bracket has been removed. 

General comments on the design: 

The authors state “The goal of this study was to measure the mitoPO2 intraoperatively under stable hemodynamic conditions. Therefore, patients were excluded if there was hemodynamic instability during surgery namely, if blood transfusion was required, if the mean arterial pressure (MAP) decreased more than 25% from baseline or if high doses of vasopressors were needed (noradrenaline ≥ 0,10 mcg/kg/min or the equivalent dose of phenylephrine)”. As outlined in the discussion (L323ff) mitoPO2 was sensitive to blood loss. I think it would be very interesting to analyse the mitoPO2 behaviour in patients with the above-mentioned changes instead of excluding them. Please comment. 

Answer: The authors agree with the reviewer that measuring mitochondrial oxygenation in unstable hemodynamic conditions would be of interest. For example in patients with major blood loss. However, the primary aim of this study was to investigate the mitoPO2 in stable conditions, as it is the first time mitoPO2 was measured during surgery in adults. Seen as we do agree that unstable hemodynamic conditions are important as well, we are currently conducting measurements during major abdominal surgery in which major blood loss is expected. In a previous version of the manuscript, we had added two example cases from the previous mentioned population. However, these cases were removed after the editor’s comments. We have now added the two cases as a supplementary file. 

The authors collected information on oxygenation with multiple instruments (O2C and other covariates). Why are no associative analysis (e.g. mixed model approach) reported? This would significantly enhance the message of the manuscript. Especially regarding the suspected association of mitoPO2 and probe temperature. 

Please consider the reporting of a statistic for stability/variability e.g. coefficient of variation. 

Answer: The authors agree with the reviewer that an associative analysis would be an important addition to the manuscript. Therefore, we have performed a mixed linear model analysis and added this to the manuscript. Furthermore, we have added a coefficient of variation in the text. 

Results 

Minor: 

L184: On average, the ALA patch was applied 17 ± 3.3 hours before surgery > this is rather long, please comment. 

Answer: The rather long application time is a result of practical reasons. Since the majority of the operations start at eight a.m., an application time of four hours would be in the middle of the night. In our other studies, including a cardiothoracic surgery population, COVID-19 patients and healthy volunteers, longer application times have proven to be effective. Therefore , we have chosen to place the plaster the afternoon or evening before the surgery [1]. 

We have added an explanation on application time in the discussion section. 

Can the authors please provide at least mean signal quality with SD and/or 95%CI at the beginning and the end of the COMET measurements?

Answer: We have added mean signal quality with standard deviation at the beginning and end of the measurements to the results. 

Additionally, we have added a paragraph on signal quality in the discussion. As after careful consideration, we have included the data of measurements with a signal quality under 20%. This is in contrast to the text in our study protocol and in the earlier version of this manuscript. We have had the possibility to analyse the data of the delayed fluorescence signal of the mitoPO2 data directly. When doing this, we learned that a signal quality of <20% as displayed by the COMET monitor does not necessarily reflect a signal to noise ratio of over 20. Therefore, with the current calculation of signal quality within the COMET, we see value in assessing the reliability of mitoPO¬2 measurements based on the raw data, instead of excluding data points based on the % shown on the COMET monitor. 

Tables 

What is the rational of reporting dexamethasone use? 

Answer: The rational for reporting dexamethasone use was the potential effect of dexamethasone on mitochondrial function. However, after consideration we have decided to leave it out of the table.

Figures 

What does the N refer to? The number of patients for the obtained means and SDs? This should be stated in the figure legends. 

Answer: The N stands for the number of patients for the obtained means and SDs, we have added this to the figure legends. 

There might be confounding between the first data points with N=20 and last data points N=10, if individual patients start from different mitoPO2 levels. Might the decrease of mitoPO2 during the first hour go back to specific individuals? E.g. in Figure 3 (B) the mitoPO2 seems rather stable. 

Answer: In the figure below we have plotted the mitoPO2 of every single patient separately. As can be seen, there is an overall decrease in all patients, not necessarily caused by specific individuals. 

Figure 1. MitoPO2 ¬during neurosurgery. The lines represent the mitochondrial oxygenation (mitoPO2 ) of 20 individual patients. 

Discussion 

L271: Please consider adding the reference. 

Answer: The reference has been added. 

L287-289: On average the mitoPO2 dropped from ~60 to ~40 mmHg. The probe temperature (and the skin temperature) increased from 34 to 35°C. The authors state that the decline in mitoPO2 most likely is not clinically relevant. 

What would be a clinical relevant change? 

Answer: In examples we have seen in ongoing trials, a mitoPO¬2 below 20 mmHg seems to be clinically relevant. However, as it was not within the objectives of this study to investigate the significance of low mitoPO2 we cannot draw any definitive conclusions. Currently we are performing several studies (both in animal models as in humans) with the objective to study clinical relevant changes. 

The authors give a potential explanation for the association of skin temperature and mitoPO2. Is it reasonable that a rather small change in skin temperature is associated with a rather “large” change in mitoPO2? 

Answer: In order to further analyze the effect of skin temperature on mitoPO2 we have performed a linear mixed model analysis. In the model with only skin temperature as predictor variable, the estimate of the effect of skin temperature on mitoPO2 was -1.76. In the model with several other hemodynamic covariates, the effect of skin temperature on mitoPO2 was -2.76. However, we hypothesize that these changes are not large. As a it means that if a patient’s skin temperature is increased with 3 degrees, the mitoPO2 is lowered with 9 mmHg, which we do not deem clinically relevant. However such temperature swings are unlikely in procedures, unless the patients are actively cooled and re-heated. 

The association of probe/skin temperature could be tested using a mixed models approach. In addition, this approach could provide adjusted mitoPO2 values. If the negative trend over time in mitoPO2 is still present, other potential covariates should be discussed. 

Answer: In light of the reviewers suggestions, we have performed a mixed linear model including several other covariates and added the results to the manuscript. 

Was there only an increase in skin temperature or also in body temperature? 

Answer: Next to a change in skin temperature, a slight change in body temperature is also expected. The increase in skin temperature is presumably caused by the hot air blanket, which covers the patient during surgery. The blanket heats the skin with 43 degrees. Therefore the increase in skin temperature is believed to be higher than the expected increase in body temperature. 

L317: “and” value of […]? 

Answer: The letter ‘d’ has been added after ‘an’ in line 317.

L323: Please consider adding the reference. 

Answer: The reference was added to line 323. 

L348: What is the meaning of “evidence based monitoring of mitoPO2”? 

Answer: We have rephrased line 348. 

L353: It should be stated that this was intraoperatively. Other studies reported COMET-measurements in healthy controls (hemodynamically stable).

Answer: We have added ‘intraoperative’ to the sentence.

L355: Only neurosurgical patients should be added. 

Answer: We have added ‘only measured in neurosurgical patients’ to the limitations.

Additional topics, which could be addressed: 

Is the skin mitoPO2 a good surrogate parameter for the central or general oxygenation? 

Answer: No, we do not think cutaneous mitoPO2 is a surrogate parameter for general oxygenation of the body. Our hypothesis is that, due to physiological compensation mechanisms, the skin is the first organ to suffer from detrimental physiological alterations (hypoxia, anemia, low cardiac output etc.) and therefore can serve as a ‘canary’ of the body. In our current thinking mitoPO2 will not be a replacement for e.g. hemoglobin saturation measurements. We hope to develop an early indicator of physiologic decompensation that is complimentary to existing clinical means of accessing oxygenation status. 

Only neurosurgical patients. What other body area would be feasible e.g. for cardiac surgery? 

Answer: Recently, our group has published a study in cardiothoracic surgery patients. In this study, measurements were performed on the upper arm. We believe that this is a good alternative, because it is also centrally located. 

Would the authors expect different mitoPO2 values in different body regions? 

Answer: In principle if all skin would have the same temperature, we would not expect a difference. However, as seen in this study, temperature has an effect of mitoPO2. Therefore, we expect the more distal regions, like hands and feet to have a different mitoPO2 value.

General Comment 

I see two options for this manuscript: 

I: If the primary aim of the manuscript is to show the feasibility and applicability of COMET-measurements during (neuro-) surgery, some of the reported results could be shortened (e.g. O2C measurements). In addition, more results on signal quality changes, excluded data

II: The manuscript could additionally focus on associative analyses. The great advantage of the design is the parallel assessment of mitoPO2, O2C variables and other covariates. Nonetheless, I’m aware that this would switch the focus of the current manuscript and would be very time consuming. 

Answer: After careful consideration, we have chosen to perform the additional analysis and have added this to the manuscript. We want to thank the reviewer for this suggestion.

Reviewer #3: 

Major Comments:

There is no discussion of the validity of the mitoPO2 monitor. What the face validity of this monitoring device is, in other words, if this device measures what it claims to measure, mitochondrial oxygenation, is not discussed. This discussion needs to include preclinical studies that are foundational for the development of this monitor. For example, are there animal studies performed to show the validity of this measurement? Was this measurement compared to other types of the mitoPO2 measurement or any existing related monitors? Content validity, whether this measurement covers all of the thing the authors are trying to measure, may deserve a consideration, too. If the device has not been tested for the validity, feasibility testing seems to be useless.

Answer: The PpIX-TLST has been in development from 2006 and has been subject to a number of reviews covering many of the above mentioned questions [2, 3]. We understand it is interesting for the PLOS ONE readers and have added a short paragraph on calibration and validation in the introduction. 

A major scientific question is the role of mitoPO2. A possible role of mitoPO2 may be an early detection of tissue/cellular hypoxia before currently existing hemodynamic and oxygen monitors can do so. If that is the question, I believe that large animal studies, such as pigs, can answer the question in much more controlled conditions. Are these studies done in animals?

Answer: The reviewer is correct and the suggested possible role of mitoPO2 is exactly what we are aiming at. And yes, the possible role of mitoPO2 is subject of large animal studies. In the study by Römers et al. [4], the mitoPO2 decreased before the mean arterial pressure and serum lactate during a porcine hemodilution protocol. Follow up animal studies are under way, as our focus is on translational research. The lessons (and questions) learned in our ongoing clinical work will be subject of investigation in the laboratory.

Related to the above question, characteristics of the measurement needs to be explained. What is the temporal resolution of this measurement? Can the measurement reflect the mitoPO2 change at the second, minute or longer-level? How is the 5-min interval determined? The temporal resolution is important for interpreting the data and the usefulness as a monitor. In addition, how does the skin mitoPO2 reflect major organ mitoPO2? Are they correlated? If there are discrepancies between skin and other organs, the monitoring of the skin mitoPO2 may not be helpful. Further, the authors discuss that the decline of mitoPO2 during surgery may be associated with the skin temperature change. How the skin and ambient temperatures and warming methods affect the mitoPO2 should be easily tested on animals.

Answer: A single mitoPO2 measurement takes up to 6 milliseconds. The technical implementation in the COMET allows for one measurement every second, but in our laboratory setups we can measure up to 20 Hz (the limitation here is the laser repetition rate).

In light of the question of the correlation between skin mitoPO2 and other organs please see our answer to reviewer 2:

Answer: This is a valid and intriguing question. We have previously measured the mitoPO2 in different organs as well as the skin. The exact mitoPO2 values were differing between the locations, but the mitoPO2 showed the same behavior in all measurement sites after LPS administration [5]. We are currently investigating the canary function of the skin for other organs in a study with porcine models.’

To test the effect of warming methods on mitoPO2 is a good suggestion. We have not tested this as we were unaware of this effect. In future animal studies we will include the above mentioned suggestion as a research objective. 

This study measured mitoPO2 in the limited time period (stable intraoperative) and in the limited patients. Anesthesia induction and emergence, hemodynamic instability, blood transfusion, moderate – high dose vasopressors are all excluded. Patients with presence of mitochondrial diseases, pregnancy or lactation are also excluded. However, the authors discuss a case report in which the mitoPO2 measurement was useful in a patient who had a large amount of blood loss. Therefore, it is not clear to this reviewer why the authors excluded all the instabilities, which seems to be a missed opportunity.

Answer: Please also see the answer to the question of reviewer 2: ‘The authors agree with the reviewer that measuring mitochondrial oxygenation in unstable hemodynamic conditions would be of interest. For example in patients with major blood loss. However, the primary aim of this study was to investigate the mitoPO2 in stable conditions, as it is the first time mitoPO2 was measured during surgery in adults. Seen as we do agree that unstable hemodynamic conditions are important as well, we are currently conducting measurements during major abdominal surgery in which major blood loss is expected. In a previous version, we had added two example cases from the previous mentioned population. However, these cases were removed after the editor’s comments. We have now added the two cases as a supplementary file.

Other Comments:

Are there any possible adverse effects of Alacare? Is Alacare absorbed systemically? Is Alacare approved for clinical use in the European nations? 

Answer: Currently topical Alacare is approved for the treatment of actinic keratosis. Systemic Alacare is approved for visualisation of malignant gliomas during neurosurgery. After local application there is no clinically relevant systemic absorption. The possible adverse effects of Alacare include: itching, a burning feeling/pain, erythema, irritation, scab formation, exfoliation on the treated skin, hyper/hypopigmentation, blister formation, pustules, erosion, edema, bleeding of the treated skin and headache. 

Line100: “Hypertension was medically induced during the hemostasis phase”. Is this induced hypertension for neurosurgical procedures?

Answer: Yes, in our hospital it is standard procedure to medically induce hypertension during the hemostasis phase for neurosurgical procedures.

Line 171: What are the causes of low signal quality?

Answer: A low signal quality is caused by a low signal to noise ratio of the optical signal. Potential reasons for this phenomenon include: interference of background light, pressure on the probe and insufficient uptake of aminolevulinic acid in the skin. 

Line 198: When exactly the baseline mitoPO2 is measured? Is this before or after surgery started. Why is the awake mitoPO2 not included? 

Answer: The patients are measured at the start of surgery after the induction of anaesthesia. This was due to two main factors, the first logistical and the second is the influence of ambient light. Thus only after the measurement site was covered from direct light sources by surgical covers could the measurement be conducted. Based on these experiences a sensor holder has been developed that covers the measurement site and enables the measurements in ambient lighting. In future studies it will therefore be possible to include peri- and postoperative measurements.

Line 276: The effect of anaesthesia should be tested in preclinical animal models. 

Answer: Thank you for your suggestion. We are indeed considering to study the effects of anesthesia in a preclinical model. Also, we are currently setting up a clinical study in which we aim to analyze the effects of deep procedural sedation. 

Figure 1 and 2: What are N20, N16, etc.? The averaged data across patients may not be helpful as all the surgical stimulation cannot be controlled. Are the data normalized or not normalized?

Answer: The N stands for the number of patients for the obtained means and SDs, we have added this to the figure legends.We appreciate the concern and have therefore added a mixed linear model which addresses the uncontrolled variables. Moreover, the data is not normalized.

Diagram for the mitoPO2 measurement mechanism may be helpful for the readers. 

Answer: We have added an explanatory figure to the methods. 

1. Streng L, de Wijs CJ, Raat NJH, Specht PAC, Sneiders D, van der Kaaij M, et al. In Vivo and Ex Vivo Mitochondrial Function in COVID-19 Patients on the Intensive Care Unit. Biomedicines. 2022;10(7). Epub 2022/07/28. doi: 10.3390/biomedicines10071746. PubMed PMID: 35885051; PubMed Central PMCID: PMCPMC9313105.

2. Mik EG. Special article: measuring mitochondrial oxygen tension: from basic principles to application in humans. Anesth Analg. 2013;117(4):834-46. Epub 2013/04/18. doi: 10.1213/ANE.0b013e31828f29da. PubMed PMID: 23592604.

3. Mik EG, Balestra GM, Harms FA. Monitoring mitochondrial PO2: the next step. Current opinion in critical care. 2020;26(3):289-95. Epub 2020/04/30. doi: 10.1097/MCC.0000000000000719. PubMed PMID: 32348095.

4. Römers LH, Bakker C, Dollée N, Hoeks SE, Lima A, Raat NJ, et al. Cutaneous Mitochondrial PO2, but Not Tissue Oxygen Saturation, Is an Early Indicator of the Physiologic Limit of Hemodilution in the Pig. Anesthesiology. 2016;125(1):124-32. PubMed PMID: 27176212.

5. Harms FA, Bodmer SI, Raat NJ, Mik EG. Cutaneous mitochondrial respirometry: non-invasive monitoring of mitochondrial function. J Clin Monit Comput. 2015;29(4):509-19. Epub 2014/11/13. doi: 10.1007/s10877-014-9628-9. PubMed PMID: 25388510.

---

## [Decision Letter · Decision Letter 2]

26 Aug 2022

PONE-D-21-32161R2Monitoring of mitochondrial oxygen tension in the operating theatre: an observational study with the novel COMET® monitor.PLOS ONE

Dear Dr. Harms,

Thank you for submitting your manuscript to PLOS ONE. After careful consideration, we feel that it has merit but does not fully meet PLOS ONE’s publication criteria as it currently stands. Therefore, we invite you to submit a revised version of the manuscript that addresses the points raised during the review process.

We look forward to receiving your revised manuscript.

Kind regards,

Yu-Chang Yeh, M.D., Ph.D.

Academic Editor

PLOS ONE

Journal Requirements:

Reviewers' comments:

Reviewer's Responses to Questions

**Comments to the Author**

1. If the authors have adequately addressed your comments raised in a previous round of review and you feel that this manuscript is now acceptable for publication, you may indicate that here to bypass the “Comments to the Author” section, enter your conflict of interest statement in the “Confidential to Editor” section, and submit your "Accept" recommendation.

Reviewer #2: All comments have been addressed

2. Is the manuscript technically sound, and do the data support the conclusions?

Reviewer #2: Yes

3. Has the statistical analysis been performed appropriately and rigorously? 

Reviewer #2: Yes

4. Have the authors made all data underlying the findings in their manuscript fully available?

Reviewer #2: Yes

5. Is the manuscript presented in an intelligible fashion and written in standard English?

Reviewer #2: Yes

6. Review Comments to the Author

Reviewer #2: General comment

The authors have responded in detail to my questions and comments. The quality and informative value of the manuscript was significantly improved by the inclusion of the associative analyses. I have only a few minor comments and questions. From my point of view the manuscript is suitable for publication in PLOS ONE, when the remaining minor questions/comments are addressed.

Major

None

Minor

Line 84: The study referenced under [22] was performed in healthy controls.

he study performed in the ICU-setting is the current reference [40]:

Neu C, Baumbach P, Plooij AK, Skitek K, Götze J, von Loeffelholz C, Schmidt-Winter C, Coldewey SM. Non-invasive Assessment of Mitochondrial Oxygen Metabolism in the Critically Ill Patient Using the Protoporphyrin IX-Triplet State Lifetime Technique-A Feasibility Study. Front Immunol. 2020 May 7;11:757. doi: 10.3389/fimmu.2020.00757. PMID: 32457741; PMCID: PMC7221153.

Line 164: please use lower case for „A self-adhesive“

Line 208: “correlation”

Please consider rephrasing to “association”

LMM is a form of regression analysis.

Line 255: “Correlation analysis using LMM”

Consider using the term “associative” or “regression” analysis. LMM is a form of regression analysis.

Line 386: „consequence of the changed parameters, the patient was resuscitation with red blood“

Please consider rephrasing (resuscitated?)

L343-344: “This implies that skin temperature is the cause 344 of the decrease in mitoPO2 after the start of the surgery.”

The current study design does not allow causal inferences. Please consider rephrasing (e. g. both were associated). The same holds for Lines 449-450.

Consider discussing the proportion of explained variance (marginal R²) for skin temperature. With 2.8 % the association is rather weak.

 

Statistical analyses

Lines 199-201: “For a paired t-test, a sample size of 20 patients was calculated with an assumed mean difference of 12 mmHg and standard deviation of 18 mmHg, a type I error probability of 0.05 and a power of 80%.”

What was the rational of performing a sample size calculation for a paired t-test for a mainly descriptive study? Throughout the manuscript, no results of a paired test are reported. Was the initial aim to compare mitoPO2 at the beginning vs. the (individual) end of the surgery? This information or test might be added to the results part.

Lines 210-211: In this model, the above-mentioned covariates were entered as fixed effects and subject number and time point were entered as random effects.

What was the rational of adding a random intercept for „time point“? Please comment. Can you please provide the results if “time point” is neglected (just for the purpose of the review)?

7. PLOS authors have the option to publish the peer review history of their article (what does this mean?). If published, this will include your full peer review and any attached files.

Reviewer #2: No

---

## [Author Response · Author response to Decision Letter 2]

12 Oct 2022

We hereby submit our revised manuscript “Monitoring of mitochondrial oxygen tension in the operating theatre: first experiences with the novel COMET® monitor.” We thank the reviewers for their expertise reviewing and the editor board for the opportunity to to submit a revised version of the manuscript. 

Major

None

Minor

1. Line 84: The study referenced under [22] was performed in healthy controls.

he study performed in the ICU-setting is the current reference [40]:

Neu C, Baumbach P, Plooij AK, Skitek K, Götze J, von Loeffelholz C, Schmidt-Winter C, Coldewey SM. Non-invasive Assessment of Mitochondrial Oxygen Metabolism in the Critically Ill Patient Using the Protoporphyrin IX-Triplet State Lifetime Technique-A Feasibility Study. Front Immunol. 2020 May 7;11:757. doi: 10.3389/fimmu.2020.00757. PMID: 32457741; PMCID: PMC7221153.

Answer: The reference has been changed. 

2. Line 164: please use lower case for „A self-adhesive“

Answer: The upper case A has been replaced with a lower case a.

3. Line 208: “correlation”

Please consider rephrasing to “association”

LMM is a form of regression analysis.

Answer: The sentence has been rephrased. 

4. Line 255: “Correlation analysis using LMM”

Consider using the term “associative” or “regression” analysis. LMM is a form of regression analysis.

Answer: The sentence has been rephrased. 

5. Line 386: „consequence of the changed parameters, the patient was resuscitation with red blood“

Please consider rephrasing (resuscitated?)

Answer: The sentence has been rephrased. 

6. L343-344: “This implies that skin temperature is the cause 344 of the decrease in mitoPO2 after the start of the surgery.”

The current study design does not allow causal inferences. Please consider rephrasing (e. g. both were associated). The same holds for Lines 449-450.

Answer: The sentences have been rephrased. 

7. Consider discussing the proportion of explained variance (marginal R²) for skin temperature. With 2.8 % the association is rather weak.

Answer: The following has been added to the discussion section: 

‘In the mixed model for skin temperature, with subject as random effect and time point as both random and fixed effect, the marginal R2 was 3.4 %, which suggests that only a small part of the variation in mitoPO2 can be explained by skin temperature. When all the included covariates are considered, this marginal R¬2 is 20.8% and the conditional R¬2 87.8 %. This insinuates that a large part of the variation in mitoPO2 is explained by the random effect part.’

 

Statistical analyses

8. Lines 199-201: “For a paired t-test, a sample size of 20 patients was calculated with an assumed mean difference of 12 mmHg and standard deviation of 18 mmHg, a type I error probability of 0.05 and a power of 80%.”

What was the rational of performing a sample size calculation for a paired t-test for a mainly descriptive study? Throughout the manuscript, no results of a paired test are reported. Was the initial aim to compare mitoPO2 at the beginning vs. the (individual) end of the surgery? This information or test might be added to the results part.

Answer: The following information has been added to the method section and results part: 

‘A paired t-test was used to compare the mean baseline mitoPO2 value to the mean ‘end-of-surgery’ mitoPO2 value.’ 

‘When compared to the ‘end of surgery’ mitoPO2 (M = 47, SD = 14) using a paired t-test, mean baseline mitoPO2 (M = 60, SD = 19) was significantly higher (t(19) = 3.845, p = 0.001).’

Lines 210-211: In this model, the above-mentioned covariates were entered as fixed effects and subject number and time point were entered as random effects. What was the rational of adding a random intercept for „time point“? Please comment. Can you please provide the results if “time point” is neglected (just for the purpose of the review)?

Answer: A random effect for time point was added to look at the random slopes. We have tested the difference between a model with and without a random effect for time point (p < 0.001). Based on this, we have added the random effect for time point in the model. Furthermore, we added time point as a fixed effect as well.

The reviewer asked for the results if time point is neglected, which can be seen in the figure below.

---

## [Decision Letter · Decision Letter 3]

6 Nov 2022

PONE-D-21-32161R3Monitoring of mitochondrial oxygen tension in the operating theatre: an observational study with the novel COMET® monitor.PLOS ONE

Dear Dr. Harms,

Thank you for submitting your manuscript to PLOS ONE. After careful consideration, we feel that it has merit but does not fully meet PLOS ONE’s publication criteria as it currently stands. Therefore, we invite you to submit a revised version of the manuscript that addresses the points raised during the review process.

We look forward to receiving your revised manuscript.

Kind regards,

Yu-Chang Yeh, M.D., Ph.D.

Academic Editor

PLOS ONE

Journal Requirements:

Reviewers' comments:

Reviewer's Responses to Questions

**Comments to the Author**

1. If the authors have adequately addressed your comments raised in a previous round of review and you feel that this manuscript is now acceptable for publication, you may indicate that here to bypass the “Comments to the Author” section, enter your conflict of interest statement in the “Confidential to Editor” section, and submit your "Accept" recommendation.

Reviewer #2: All comments have been addressed

2. Is the manuscript technically sound, and do the data support the conclusions?

Reviewer #2: Yes

3. Has the statistical analysis been performed appropriately and rigorously? 

Reviewer #2: Yes

4. Have the authors made all data underlying the findings in their manuscript fully available?

Reviewer #2: Yes

5. Is the manuscript presented in an intelligible fashion and written in standard English?

Reviewer #2: Yes

6. Review Comments to the Author

Reviewer #2: Maybe this is just a problem with the Word review mode:

A: "with every increase of -1 degrees Celsius in skin temperature, mitoPO2 decreased by 1.76 mmHg"

Increase of -1 is a implausible formulation

B:"Additionally, the time point of the measurement, microvascular blood flow, StO2, and SpO2 were correlated with

mitoPO2 (p < 0.001, p = 0.015, p = - 0.017 and p < 0.001"

The bracket at the end of the sentence is missing.

P-values cannot be less than 0.

C: "It should be noted that due to the small sample size, the assumption that there is no measurement error has not been corrected for."

Consider re-phrasing slightly.

7. PLOS authors have the option to publish the peer review history of their article (what does this mean?). If published, this will include your full peer review and any attached files.

Reviewer #2: No

---

## [Author Response · Author response to Decision Letter 3]

14 Nov 2022

Answers to the reviewers concerning the manuscript ‘Monitoring of mitochondrial oxygen tension in the operating theatre: an observational study with the novel COMET® monitor.’

1. Reviewer #2: Maybe this is just a problem with the Word review mode: A: "with every increase of -1 degrees Celsius in skin temperature, mitoPO2 decreased by 1.76 mmHg"

Answer: In our manuscript line 265 reads: With every increase of 1 degrees Celsius in skin temperature, mitoPO2 decreased by 1.76 mmHg’. Therefore, we believe that, as the reviewer suggested, this might be a problem with the Word review mode. 

2. B:"Additionally, the time point of the measurement, microvascular blood flow, StO2, and SpO2 were correlated with mitoPO2 (p < 0.001, p = 0.015, p = - 0.017 and p < 0.001". The bracket at the end of the sentence is missing-values cannot be less than 0.

Answer: The bracket has been added. Furthermore, the misplaced minus sign before p = 0.017 has been removed. 

3. "It should be noted that due to the small sample size, the assumption that there is no measurement error has not been corrected for." Consider re-phrasing slightly.

The sentence has been rephrased to: ‘It should be noted that measurement errors were not corrected for

---

## [Editor Report · Decision Letter 4]

21 Nov 2022

Monitoring of mitochondrial oxygen tension in the operating theatre: an observational study with the novel COMET® monitor.

PONE-D-21-32161R4

Dear Dr. Harms,

We’re pleased to inform you that your manuscript has been judged scientifically suitable for publication and will be formally accepted for publication once it meets all outstanding technical requirements.

Kind regards,

Yu-Chang Yeh, M.D., Ph.D.

Academic Editor

PLOS ONE

---

## [Editor Report · Acceptance letter]

1 Feb 2023

PONE-D-21-32161R4 

Monitoring of mitochondrial oxygen tension in the operating theatre: an observational study with the novel COMET monitor. 

Dear Dr. Harms:

I'm pleased to inform you that your manuscript has been deemed suitable for publication in PLOS ONE. Congratulations! Your manuscript is now with our production department. 

Kind regards, 

on behalf of

Dr. Yu-Chang Yeh 

Academic Editor

PLOS ONE